# Cascade Oxygen Control Enhances Growth of *Nicotiana benthamiana* Cell Cultures in Stirred-Tank Bioreactors

**DOI:** 10.3390/plants14182879

**Published:** 2025-09-16

**Authors:** Fuensanta Verdú-Navarro, Juan Antonio Moreno-Cid, Julia Weiss, Marcos Egea-Cortines

**Affiliations:** 1Bioprocess Pilot Plant, Bionet, 30320 Fuente Álamo, Spain; fuensanta.verdu@bionet.com (F.V.-N.); juan.morenocid@bionet.com (J.A.M.-C.); 2Instituto de Biotecnología Vegetal, Universidad Politécnica de Cartagena, 30202 Cartagena, Spain; julia.weiss@upct.es; 3Probelte SAU, 30500 Molina de Segura, Spain

**Keywords:** plant cell culture, Packed Cell Volume (PCV), bioprocess, specific growth rate (µ), volumetric mass transfer coefficient (*k*_L_*a*), reynolds number (NRE), agitation, aeration, scale-up

## Abstract

Plant cell cultures offer a promising platform for producing valuable biomolecules, yet their use in bioreactors remains under-optimized. Compared to animal or microbial cells, plant cells grow more slowly, limiting productivity. A common bioprocess strategy to improve yields involves controlling dissolved oxygen (DO) levels. However, little research has focused on combining agitation and aeration to regulate oxygen in plant cell cultures within bioreactors. The aim of this study was to evaluate the impact of an oxygen cascade mixing agitation and aeration on plant cell growth in stirred-tank systems. By maintaining 30% DO through this approach, the specific growth rate (µ) increased from 0.082 d^−1^ to 0.144 d^−1^ on average in *Nicotiana benthamiana* cultures at the 2 L scale, decreasing batch lengths from 21 to 10 days. These conditions were successfully replicated in a 7 L stainless-steel pilot bioreactor using previous values of geometry, *k*_L_*a* and *N*_RE_ as scale-up criteria, obtaining a µ of 0.161 d^−1^. These results demonstrate that plant cell cultures’ efficiency can be enhanced by using standard bioprocess parameters. While this work confirms the feasibility of cascade oxygen control for improvements in growth, further studies are needed to evaluate its specific impact on biomolecule production across different systems.

## 1. Introduction

Plant cell cultures in bioreactors are gaining importance in the biotechnology industry as a sustainable way to produce a wide variety of biomolecules or complex products such as meat, biopolymers or biofuels [1,2,3,4,5,6]. Plant cell cultures offer a robust chassis for the production of secondary metabolites and recombinant protein for different applications [7,8,9]. Rice cell cultures can be used to produce Fibroblast Growth Factor to propagate both human embryonic stem cells (hESCs) and pluripotent stem cells (hiPSCs), which are used in regenerative medicine [10]. Products such as triterpenes have interesting properties for pharmaceutical, nutraceutical and cosmetic applications [11,12,13,14]. Their production using apple cell suspension cultures has been achieved in bioreactors, with high yields [15]. Another example that has achieved commercial scope is the production of taxol. Two companies, Phyton Biotech Inc. and Samyang Genex Corp., are currently producing paclitaxel and related taxanes using large-scale fermentation processes [16,17]. Since 1993, Phyton has partnered with Bristol–Myers Squibb to explore the commercial potential of plant cell fermentation technology. In 2002, this collaboration expanded, with a long-term, multimillion-dollar agreement for Phyton to supply paclitaxel. Together, they operate the world’s largest plant cell fermentation facility, with a production capacity of 75,000 L.

Although rapid cell growth is important for achieving high biomolecule yields, it is not the sole determinant. Secondary metabolite production is frequently stimulated by elicitors such as salicylic acid, cyclodextrins, methyl jasmonate or yeast extract [18,19,20,21]. Nevertheless, culture growth speed is an important parameter to achieve cell densities that allow enhanced yields in biomolecule production.

There are several ways to optimize culture growth in bioreactors. One of the strategies is to improve the process parameters of the culture itself, such as agitation, aeration or temperature. One key parameter is dissolved oxygen (DO), which should be maintained at a certain level to reach a high biomass concentration in *E. coli*, yeast and CHO (Chinese hamster ovary) cells [22,23,24]. The effect of oxygen levels has also been studied in plant cell cultures. For example, low levels of DO cause decreased growth rate of *Daucus carota* cell cultures [25]. DO level is controlled by changing the aeration rate and the concentrations of oxygen, nitrogen and air in rice cultures in bioreactors [26]. Other studies use O_2_ enrichment directly to maintain the DO level at 30% during the culture, with agitation and aeration rates constant [27]. DO levels and gradients inside the bioreactor are a key parameter for the culture performance of *Nicotiana tabacum* cells [28,29,30]. This suggests that the DO level control of plant cell cultures in bioreactors with a cascade that implies both agitation and aeration rates could be a good strategy to optimize growth conditions and, therefore, production yields. When studying the effect of stirring speed on growth and production of metabolites using *Buddleja cordata* cell cultures, increased agitation speed shows a higher biomass and metabolite production [31]. However, agitation can cause cell shearing, so there is a limit to the maximum agitation that can be implemented for a given cell type [27]. DO levels are achieved by a combination of stirring and aeriation, but they are subject to changes related to the progressive increase of biomass in the bioreactor and the O_2_ consumption by the cells. Surprisingly, there are no reports describing cascade control comprising both aeration and agitation.

A recurrent challenge in bioprocessing is scaling up using predictive empirical parameters. Various scale-up strategies are used in bioreactors, typically based on maintaining the constancy of specific parameters [32]. Commonly used parameters include the volumetric mass transfer coefficient (*k*_L_*a*) [33], power input per unit volume (P/V) [34], Reynolds number (*N*_RE_), aeration rate [34] and mixing time (tm). The volumetric mass transfer coefficient (*k*_L_*a*) quantifies the rate of gas–liquid transfer (e.g., O_2_ or CO_2_), with higher values indicating a more efficient transfer. Power input per unit volume (P/V) reflects the energy delivered by the agitator to the culture broth and is a key determinant of reactors’ design and scalability. The Reynolds number (*N*_RE_) distinguishes laminar from turbulent flow, guiding shear control during scale-up. The aeration rate, expressed in vvm (volume of gas per volume of medium per minute), ensures consistent a gas supply across scales. Impeller tip speed, defined as the distance traveled per unit time by the impeller tip, enables equivalent agitation in reactors of different sizes. Mixing time, the time required for contents to reach compositional homogeneity after tracer’s addition, should be minimized to improve uniformity and process performance. The industry relies heavily on CSTRs, which provides several advantages over other systems. These include efficient mixing, operational versatility and ease of scale-up. The structural components of the bioreactor influencing cell growth include impeller design [35], aeration pore size [36] and vessel geometry [37].

In plant cell cultures, scale-up in airlift bioreactors has been achieved at volumes up to 20 L using an aeration rate expressed in vvm as the scaling criterion [38]. The *k*_L_*a* has also been used as a scale-up parameter for transgenic rice cell cultures [39]. However, the simultaneous use of multiple parameters, such as *k*_L_*a*, *N*_RE_, vvm, tip speed and bioreactor geometry, has not yet been reported for plant cell cultures.

Plant cells can be cultivated in various types of bioreactors. Airlift bioreactors, which use pneumatic agitation through gas sparging and fluid circulation, are particularly suitable for plant cell cultures that are highly sensitive to shear stress [40,41,42,43].

Another low-shear alternative is the wave or 2D rocking bioreactor, which also offers the advantage of being single-use [44,45]. Rocking bioreactors differ from stirred-type bioreactors, because aeration or oxygen input is performed on the free head-space above the culture and not by bubbling using a sparger [46]. These types of setups have been successfully used to produce crocin using *N. tabacum* and *N. benthamiana* transgenic lines [47].

The aim of this work was to determine the possibility of plant cell cultures’ optimization using a cascade to control the DO, modifying both the agitation and aeration in an STR bioreactor. We used *N. benthamiana* cell cultures, as it has been extensively used for production of biomolecules both in whole plants and cell cultures [48,49,50,51]. We scaled up the process from a 2 L bench-top bioreactor to a 7 L pilot-scale bioreactor using the constancy of *k*_L_*a* and *N*_RE_, the vvm, the tip speed and the geometry as scale-up criteria.

## 2. Results

### 2.1. Effect of DO Cascade Control by Agitation and Aeration

We performed a total of eight batches of *N. benthamiana* cell cultures in a bioreactor at bench-top scale, comprising four without cascade control and four with cascade control. The duration and the PCV (Packed Cell Volume) achieved in each batch is presented in Table 1. The batch duration in bioreactor production was significantly different in uncontrolled versus controlled batches. Batches without cascade control took between 19 to 21 days, whereas the batches with cascade control had a duration of 9 to 14 days. The final PCV achieved ranged between 70% and 90% and showed no significant differences between treatments. As expected from the shortened growth period, the specific growth rate (µ) was significantly higher in controlled versus uncontrolled batches. The growth rates ranged between 0.076–0.089 d^−1^ for uncontrolled DO and 0.11–0.173 d^−1^ under controlled DO conditions.

We took one replicate of each treatment to analyze in detail the kinetics of DO, pH, temperature, aeration and agitation. Under the uncontrolled DO conditions, DO starting at 68% decreased gradually right after inoculation. This trend was maintained until the end of the culture, lasting 21 days (Figure 1A,B). At the end of the culture, the DO level was 8.7%. The initial pH was 5.6 and decreased steadily during the first 60 h. Throughout most of the cultivation period, pH values remained within the range of 4.96 to 5.15. A sharp increase in pH was observed at the end of the growth phase, as previously found by other authors [52], coinciding with a marked rise in DO levels, both indicating cultures’ termination. Final PCVs ranged from 70% to 88% (Figure 1C).

We evaluated the effect of implementing a cascade strategy of controlled DO levels. This strategy involved progressively increasing both agitation and aeration rates to meet the DO set point. Preliminary experiments indicated that manually increasing agitation and aeration enhanced the culture’s growth rate. Based on these findings, agitation was controlled within a range of 80–160 rpm and aeration between 0.4 and 1.2 slpm (Figure 2A). The cascade was configured to prioritize agitation increases, followed by aeration once the agitation limit was reached.

Under these conditions, DO levels dropped rapidly immediately after inoculation (Figure 2B), triggering an automatic increase in agitation, which reached its maximum value within the first few hours. This drastic drop in the DO levels may be due to the inoculum used. This response stabilized DO at the set point of 30%. However, the aeration rate exhibited a stepwise increase, reaching its maximum at 114 h. Once both agitation and aeration reached their respective limits, DO levels declined below 30%, indicating increased oxygen demand by the culture.

This strategy significantly reduced the total culture time, achieving saturation in times ranging between 9 and 14 days (Table 1). The pH profile followed a similar trend to that observed in the control culture without cascade control, further confirming culture completion at day ten. Final PCV values reached approximately 90% (Figure 2C). Using the specific growth rate formula, this improvement corresponded to increasing the specific growth rate from 0.082 d^−1^ (uncontrolled DO culture) up to 0.144 d^−1^ (cascade-controlled DO culture). This represents on average a 1.76-fold increase in productivity, with a maximum difference of 2.28-fold, confirming the positive effect of applying DO cascade control to maintain oxygen levels within the optimal range during exponential growth, by automatically regulating agitation and aeration.

### 2.2. Process Scale-Up Using Reynolds Number and k_L_a Parameters

To evaluate the engineering performance of the bioreactor systems, two key scale-up parameters were analysed: *N*_RE_ and *k*_L_*a*. The *N*_RE_ provided insight into the flow regime within the vessels, revealing a laminar flow (*N*_RE_ ~ 580–910) typical of low-shear environments suitable for plant cell cultures.

Concurrently, *k*_L_*a* measurements showed that the oxygen transfer capacity could be controlled and even increased from the range obtained in the laboratory scale to the pilot scale. The conditions used for measuring the *k*_L_*a* are presented in Table 2 for each bioreactor. The measurements were performed in triplicate for each condition. At the laboratory scale, we obtained *k*_L_*a* values of 6.1 and 20 h^−1^ at the minimum and maximum ranges of the agitation and aeration, respectively. However, when we increased both ranges of the agitation and aeration, we achieved *k*_L_*a* values of 13.4 and 41.6 h^−1^, at the minimum and maximum conditions, respectively.

To clearly illustrate the engineering parameters used for scale-up and their alignment across bioreactor configurations, a comparative summary is presented in Table 3. This table includes the critical process variables for both laboratory and pilot scales, highlighting the optimized conditions achieved through DO cascade control.

### 2.3. Effect of DO Cascade Control at Larger Scale

The conditions established in the F2 bioreactor were 25 °C, 0.2 barg and a cascade strategy to control DO levels at 30%. We replicated the same strategy followed at the bench-top bioreactor. First, we increased the agitation, followed by the aeration rate. The agitation range was between 80 and 200 rpm, while the aeration ranged between 4 and 8 slpm (Figure 3A). When we analyzed the growth in the F2 bioreactor under these conditions, we observed that the DO level decreased rapidly after inoculation (Figure 3B). However, the DO level decreased at a slower pace than observed in the bench-top bioreactor. This was due to the starting conditions established, as they corresponded to a higher oxygen transfer rate, since there were higher agitation and aeration rates. It decreased below 30% after 150 h. At this point, the agitation rate started to increase gradually to maintain the level at 30%, reaching its maximum at 190 h. The aeration followed the cascade control once agitation reached its maximum. At one point (hour 210), there was an increase in air input, causing an increase in DO above 30% that caused a decrease in agitation. The pH profile of the culture was very similar to those from the bench-top bioreactor cultures. It also showed that the culture had reached saturation after 10 days. When we harvested the culture, we found that PCV had reached 80% (Figure 3C). Using this final PCV and the duration of the culture, we calculated the specific growth rate, and it was 0.161 d^−1^. This demonstrated that the use of a cascade strategy to improve bioprocesses in plant cell cultures is reproducible at larger scales.

## 3. Discussion

Plant cell platforms are emerging as valuable systems for producing secondary metabolites, proteins and high-value biomolecules for the pharmaceutical industry. However, compared to microbial (e.g., *E. coli*, *Komagataella phaffii*) or mammalian (e.g., CHO) platforms, knowledge of plant cell bioprocesses in bioreactors remains limited. While operational parameters such as temperature and medium composition differ between cell types, general principles for bioreactor cultivation can be established. Among the most critical factors in bioreactor cultures are oxygen availability, shear sensitivity, pH regulation and inoculation strategies. Inoculum volume strongly influences plant cell cultures’ performance [53,54], though it was not varied in this study. Similarly, pH is a determinant of plant cell development and metabolite production, influencing processes such as somatic embryogenesis and crocin biosynthesis in CSTRs [55,56,57]. In the present work, we investigated the effects of aeration and agitation on *Nicotiana benthamiana* cell growth in a CSTR, with pH monitored but not controlled.

### 3.1. Effect of DO Control on Culture Performance

DO is a key parameter influencing cell growth across microbial, mammalian and plant systems [22,23,24,58]. Low DO levels are associated with reduced growth rates in multiple plant species, including *Daucus carota* [25], *Catharanthus roseus* [59,60] and *N. tabacum* [30]. Beyond growth, DO also impacts metabolite biosynthesis, such as phenolic compounds in tobacco [30], rosmarinic acid in *Lavandula vera* [61] or paclitaxel production in *Taxus chinensis* [62], with studies conducted in both shake flasks and CSTRs. The growing adoption of wave bioreactors has introduced alternative DO regulation strategies, including modulation of viscosity, application of antifoam [63], and surface aeration [64], which consistently maintain DO near 50%, a condition generally favorable for plant cell proliferation.

In our uncontrolled system, DO values frequently fell below 30%, likely inducing oxygen limitations and delaying culture maturation. By contrast, cascade-controlled operation provided a more stable oxygen supply, sustaining aerobic metabolism and accelerating accumulation of biomass. The increase in specific growth rate (μ) from 0.082 d^−1^ to 0.144 d^−1^ under controlled conditions highlights the critical role of DO in plant cell bioprocesses. Previous reports on transgenic *N. benthamiana* lines documented μ values of 0.113 d^−1^ in shake flasks and 0.14 d^−1^ in bioreactors [48,65], with a further improvement to 0.26 d^−1^ when DO was maintained at 40% through high aeration and oxygen supplementation [65]. In our study, DO cascade control enhanced μ from 0.082 d^−1^ to 0.144 d^−1^ at laboratory scale and up to 0.161 d^−1^ at pilot scale. Although these rates are slightly lower than the reported values [65], the difference is likely attributable to our lower DO set point. Importantly, our approach combined aeration and agitation to achieve fine-tuned DO control, suggesting that further improvements may be attainable by raising the DO set point and incorporating oxygen enrichment.

### 3.2. Scaling Up the Process with N_RE_, k_L_a and Geometry

The observed Reynolds numbers in both scales fell within the laminar flow regime (*N*_RE_ ~ 200–1000), characteristic of plant cell cultures where low agitation rates are enforced to minimize shear stress. Although the pilot-scale system presented slightly higher *N*_RE_ values at maximum agitation and viscosity, the order of magnitude remained comparable between scales, which implies similar hydrodynamic regimes. Indeed, work on *Sphaeralcea angustifolia* shows that *N*_RE_ is a critical parameter, and increased values beyond a laminar flow regime cause cell death and decreased production [66], indicating that maintaining the *N*_RE_ values within those limits is critical.

Given the limitations of using *N*_RE_ alone for scale-up in plant cell cultures, due to their non-Newtonian and shear-sensitive nature, the process was scaled using the *k*_L_*a* as the primary criterion and the tip speed and vvm, since they impact the *k*_L_*a* values. Target *k*_L_*a* values of 6.1 to 20.05 h^−1^ were met at both scales and even increased in the pilot scale up to 13.4 and 41.6 h^−1^, respectively. These findings are consistent with previous reports where *k*_L_*a* values in *Nicotiana tabacum* and *Thalictrum minus* were maintained at 12 h^−1^ and 6.2 h^−1^, respectively [28]. However, *k*_L_*a* values as low as 1.8 h^−1^ has been measured in cultures of *Bacopa monnieri,* producing bacosides, indicating that the specific *k*_L_*a* value is dependent on the cell type [67]. A *k*_L_*a*-based scale-up strategy ensures a sufficient oxygen supply across different scales, while preserving a comparable mixing efficiency and low-shear conditions. This approach is especially relevant for the optimization strategy presented in this study, which maintains DO levels at 30%.

These results support the validity of using *k*_L_*a* as a reliable scale-up criterion for suspended plant cell cultures, enabling consistent oxygen availability without exceeding agitation levels that could harm the cells. Additionally, the combined analysis of *N*_RE_ and *k*_L_*a* data confirms that similar mixing and mass transfer conditions were achieved across scales, demonstrating the robustness and scalability of the process.

Given these results, it is important to consider their industrial implications. The engineering strategy demonstrated here is directly translatable to larger-scale systems exceeding 50 L. Indeed, several studies, including an up-scale from 15 mL to 2000 L and down-scale from 2000 L to 250 mL, show that using *k*_L_*a* as scale-up criteria is feasible and reproducible [68,69,70]. However, bioreactors with conserved geometry may play a critical role in maintaining essential oxygen transfer parameters, such as *k*_L_*a* and *N*_RE_, an area that warrants further investigation. Notably, the pilot-scale *k*_L_*a* values achieved (up to 41.6 h^−1^) fall within the operational range of larger single-use or stainless-steel stirred-tank bioreactors commonly used in industrial bioprocessing [71,72,73,74].

Furthermore, the average µ (0.144 d^−1^) and the reduced culture duration (10 days) indicate strong potential for intensification of the process. Given the moderate shear environment (*N*_RE_ ≈ 900), these conditions are likely compatible with shear-sensitive plant cell lines and could be further scaled up to larger volumes, with appropriate adjustments to gas flow control and impeller design.

Our work has practical implications, as it shows a protocol to improve growth rate and scalability using plant cell cultures in a CSTR.

## 4. Materials and Methods

### 4.1. Plant Material

We obtained plant cell cultures from the LAB strain of *Nicotiana benthamiana* [75]. Seeds were surface-sterilized with a solution of commercial bleach at 50% and 0.2% Tween20 for 10 min and then rinsed with sterile water four times. Seeds were germinated in square Petri dishes with ½ MS medium [76], 10 g/L sucrose and 8 g/L Phyto agar. Germinated seeds were transferred to Steri Vent containers with the same medium as described above. The growth conditions were a 16/8 light/darkness photoperiod, with a thermoperiod of 23/18 °C, respectively.

### 4.2. Callus Culture

Leaf explants were excised from full grown in vitro plants. They were cultured in Petri dishes with a callus induction medium consisting of 4.3 g/L MS medium, 30 g/L sucrose, 0.1 g/L myo-inositol, 0.204 g/L KH_2_PO_4_, 0.5 mg/L nicotinic acid, 0.5 mg/L pyridoxine, 0.5 mg/L thiamine, 0.4 mg/L 2,4-dichlorophenoxyacetic acid, 0.1 mg/L kinetin and 7.5 g/L Phyto agar. Culture conditions were continuous light and 23 °C. After four weeks, calli were transferred to Steri Vent containers with the same medium. Subculture was performed every three weeks until they were friable and could be disaggregated in liquid medium. Enough friability was achieved after three months, approximately.

### 4.3. Shake Flask Cultures

After 3–4 weeks, friable calli were selected and transferred to flasks with liquid medium to initiate suspension cultures. The liquid cell culture medium’s composition was described previously [77]. The medium pH was adjusted to 5.8 prior sterilization at 121 °C for 30 min. Cell cultures were inoculated with friable calli at a 20% rate to assure cell viability. Cell suspensions were maintained in 500 mL Erlenmeyer flasks with 200 mL liquid medium. Flasks were placed in an orbital shaker at 110–120 rpm (Stuart orbital Shaker-SSL1, Vernon Hills, IL, USA). The orbital shaker was placed in a growth chamber at 23 °C with continuous light. Subcultures were performed at 10–15-day intervals, maintaining the same inoculum rate. Log-phase suspensions were used as the inoculum for bioreactor-scale runs, with Packed Cell Volume adjusted to 16%. An inoculum propagation ratio of 20% *v*/*v* was maintained across all scales. After inoculation, cultures were allowed to grow until reaching a final Packed Cell Volume of 80% prior to further scale transfer.

### 4.4. Bioreactor Configuration and Scale-Up Strategy

Cell suspension cultures of *N. benthamiana* were cultivated in CSTR at two different scales, with the following working volumes: 2 L (laboratory scale) and 7 L (pilot scale) (Figure 4). The medium used had the same composition as the flasks’ medium. The medium was prepared and sterilized in the bioreactor vessel with probes and tubes at 121 °C for 30 min. At laboratory scale, the vessel with the medium, probes and tubes was autoclaved. At pilot scale, the bioreactor was sterilized in place (SIP) with the medium, probes and tubes. The inoculum rate was maintained at 20% *v*/*v*. Cultures were initiated with a PCV of 16% and grown until the complete depletion of the culture medium carbon source. The duration of the cultures depended on the strategy for the DO. The agitation speed and vessel configurations were chosen to reflect operational limitations typical of plant cell cultures, particularly concerning shear sensitivity.

The inoculum propagation ratio was fixed at 20% (*v*/*v*) in all cases, ensuring uniform starting conditions. PCV-based monitoring proved to be a reliable indicator of biomass accumulation, especially for shear-sensitive plant cell cultures where standard optical density measurements may be misleading.

The laboratory-scale system consisted of a Bionet F0 cell culture bioreactor (Bionet, Fuente Álamo, Murcia, Spain) equipped with a pitched-blade turbine of 54 mm diameter. The pilot-scale system used a Bionet F2 bioreactor with a 70.5 mm diameter pitched-blade impeller. The impellers both had 3 blades with an angle of 45°. Both systems were maintained at 25 °C and 0.2 bar overpressure. The DO was kept within 30% saturation by means of a cascade control strategy, which automatically adjusted agitation speed and aeration rate, using air. The F0 operated at an initial agitation of 80 rpm, with a maximum of 160 rpm. The equivalent impeller tip speed would be 0.23 and 0.45 m/s, respectively. Initial aeration started at 0.2 vvm air, with a maximum of 0.6 vvm air. The F2 operated between 80 rpm and 220 rpm. The calculated tip speed would be 0.3 and 0.82 m/s. Airflow rates used at this scale ranged between 0.6 and 1.1 vvm. This control was implemented using ROSITA 0.9.2 software for the laboratory-scale Bionet F0 bioreactor and MARTA 6.1.0 software for the pilot-scale Bionet F2 system. The pH was monitored throughout the culture but was not controlled.

The maximum working volume of the Bionet F2-15MB pilot bioreactor was estimated by preserving the height-to-diameter (H/D) ratio established in the Bionet F0-2CC bench-top unit. While the F2-15MB was originally designed with geometry optimized for microbial cultures, the F0-2CC features a configuration suited for cell cultures.

The bench-top bioreactor F0-2CC exhibited an H/D ratio of 1.15, calculated from the liquid column height at maximum working volume relative to the vessel diameter. To maintain geometric similarity during scale-up, this ratio was applied to the pilot-scale bioreactor F2-15MB, which has a vessel diameter of 211.6 mm. Using this ratio, the corresponding maximum working height was determined, resulting in a total working volume of 7 L for the pilot unit.

This geometrically consistent scaling approach ensured preservation of fluid distribution patterns between scales. To further evaluate the impact of geometric consistency on bioprocess performance, scale-up was carried out under constant *k*_L_*a* and *N*_RE_ conditions. This strategy aimed to assess whether maintaining both physical geometry and hydrodynamic conditions would sustain the high productivity previously achieved at the laboratory scale.

### 4.5. Specific Growth Rate Calculation

The specific growth rate (μ, d^−1^) was estimated from the exponential increase in PCV using the following equation:(1)$$µ\;=\;\frac{\ln\left({PCV}_{2}\right)-\;\ln\left({PCV}_{1}\right)\;}{t_{2}-\;t_{1}},$$
where PCV_1_ and PCV_2_ represent the Packed Cell Volume (%) at times t_1_ and t_2_, respectively (expressed in days). For this study, t_1_ corresponded to cultures’ initiation (PCV = 16%), and t_2_ was recorded when cultures were harvested. This parameter was used to quantify biomass accumulation efficiency under different bioreactor conditions. The PCV was determined by letting the harvested cultures sediment for 24 h at 4 °C.

### 4.6. Determination of k_L_a via Sodium Sulphite Method

The volumetric oxygen transfer coefficient (*k*_L_*a*) describes the rate at which oxygen is transferred from the gas phase into the liquid phase per unit volume. It is related to the oxygen transfer rate (OTR) by the following expression:(2)$$OTR=k_{L}a\times \left(C*-C_{L}\right),$$
where
OTR is the oxygen transfer rate (mol·L^−1^·h^−1^);*k*_L_*a* is the volumetric mass transfer coefficient (h^−1^);C* is the saturation concentration of oxygen in the medium (mol·L^−1^);C_L_ is the actual dissolved oxygen concentration in the medium (mol·L^−1^).

The sodium sulphite method is a deoxygenation-based chemical technique widely used to estimate oxygen transfer rates in bioreactors under non-biological conditions [78]. A sulphite solution (typically 0.5 mol/L Na_2_SO_3_) is prepared in deionized water, with copper sulphate (CuSO_4_) added as a catalyst at 0.001 mol/L. Upon aeration, the reaction consumes dissolved oxygen as follows:(3)$$2{Na}_{2}SO_{3}\;+\;O_{2}\to 2{Na}_{2}SO_{4}$$

The OTR was estimated based on the duration of the sodium sulphite oxidation reaction, specifically measuring the time during which the DO concentration remained below 15%. This approach leverages the stoichiometric equivalence between sodium sulphite oxidation and oxygen uptake. The rapid reaction kinetics ensure a consistently low DO level during the measurement, allowing for a reliable determination of the OTR from the observed depletion profile.

### 4.7. Reynolds Number Calculation

Viscosity values were estimated based on literature reports for plant cell suspensions with similar PCV ranges, with initial viscosity (μ) at 0.0156 Pa·s (16% PCV) and final viscosity at 0.0182 Pa·s (80% PCV) [79]. *N*_RE_ was calculated using the classical relation for stirred systems:(4)$$N_{RE}\;=\;\left(\rho \;\times N\;\times D^{2}\right)/µ,$$
where
*ρ* is the fluid density (assumed 1000 kg/m^3^);N is the impeller speed (rev/s);D is the impeller diameter (m);μ is the dynamic viscosity (Pa·s).

### 4.8. Data Analysis and Statistics

Data obtained from ROSITA and MARTA software was downloaded as .csv files. Images were created using ggplot2 [80]. Statistical analyses were performed in R version 4.3.2.

## 5. Conclusions

This study provides the first evidence that cascade control, combining aeration and agitation, can substantially accelerate plant cell growth in stirred bioreactors. Maintaining DO at 30% reduced culture times from 21 to 10 days, with results successfully reproduced at pilot scale. These findings underscore the potential of cascade-based DO regulation as a robust and scalable strategy to shorten production timelines in plant cell bioprocesses. However, the specific effects of this approach on the biosynthesis of biopharmaceuticals, vaccine components or high-value metabolites remain to be determined and are likely process-dependent. Future investigations should evaluate whether the growth-optimized conditions identified here also support enhanced metabolite and protein production.

## Figures and Tables

**Figure 1 plants-14-02879-f001:**
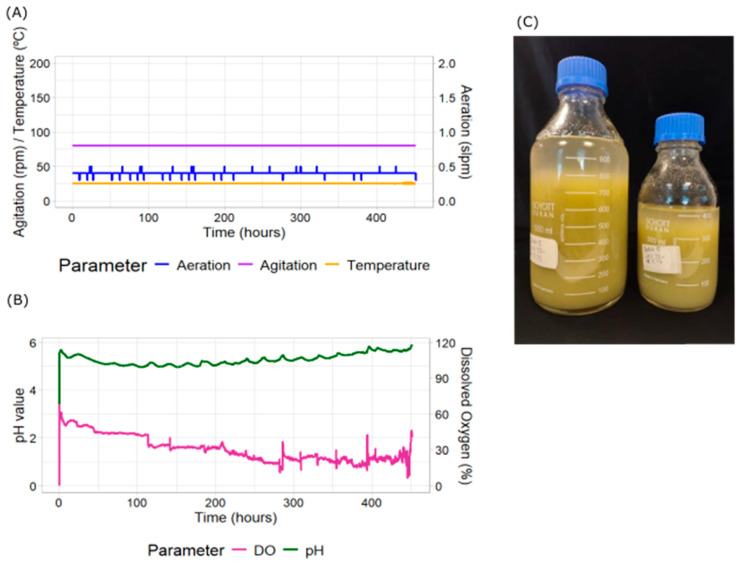
Measured parameters in *N. benthamiana* cell cultures and Packed Cell Volume (PCV) using a laboratory-scale bioreactor without cascade control. The bioreactor parameters were measured continuously during growth. These include (**A**) aeration (blue), agitation (purple) and temperature (yellow); (**B**) dissolved oxygen (pink) and pH (green). (**C**) Total biomass production as seen by PCV at the end of the culture measured after 24 h sedimentation of the total culture harvested into two bottles.

**Figure 2 plants-14-02879-f002:**
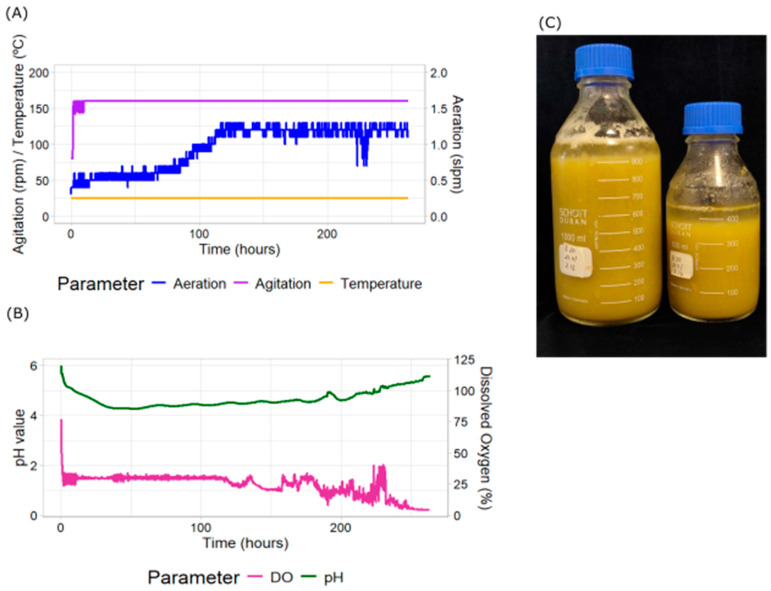
Measured parameters in *N. benthamiana* cell cultures and Packed Cell Volume (PCV) using a laboratory-scale bioreactor with cascade control. The bioreactor parameters were measured continuously during growth. These include (**A**) aeration (blue), agitation (purple) and temperature (yellow); (**B**) dissolved oxygen (pink) and pH (green). (**C**) Total biomass production as seen by PCV at the end of the culture measured after 24 h sedimentation of the total culture harvested into two bottles.

**Figure 3 plants-14-02879-f003:**
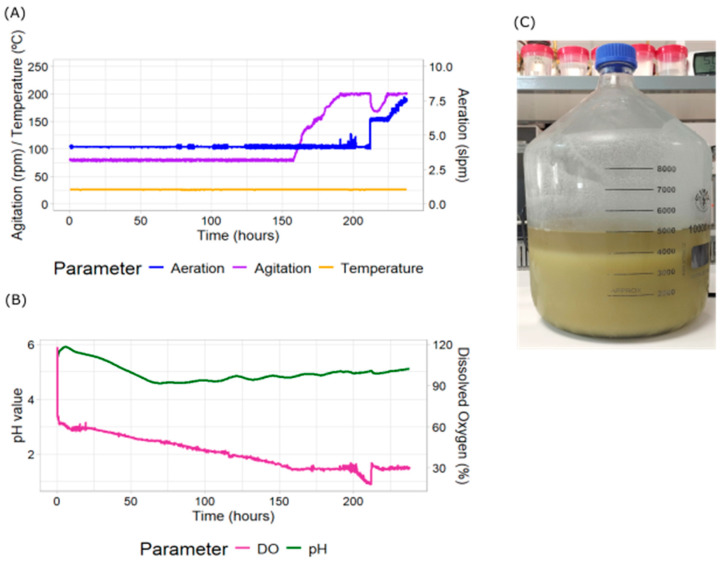
Measured parameters in *N. benthamiana* cell cultures and Packed Cell Volume (PCV) using a pilot-scale bioreactor with cascade control. The bioreactor parameters were measured continuously during growth. These include (**A**) aeration (blue), agitation (purple) and temperature (yellow); (**B**) dissolved oxygen (pink) and pH (green). (**C**) Total biomass production as seen by PCV at the end of the culture measured after 24 h sedimentation of the total culture harvested.

**Figure 4 plants-14-02879-f004:**
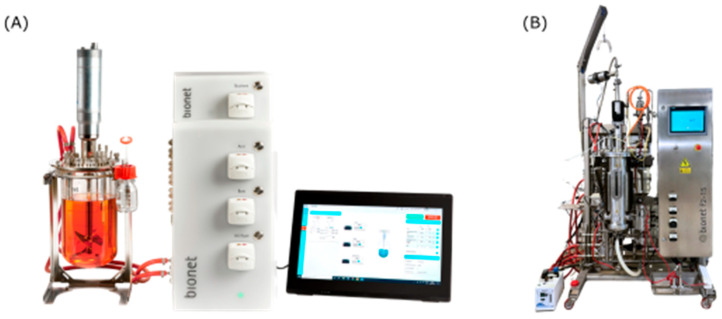
Bioreactors used for experiments: (**A**) laboratory-scale bioreactor Bionet F0-2CC and (**B**) pilot-scale bioreactor Bionet F2-15MB.

**Table 1 plants-14-02879-t001:** Overall batch duration, Packed Cell Volume (PCV) and specific growth rate (µ) of eight batches in laboratory-scale F0-2CC bioreactor. Four replicates were run with uncontrolled dissolved oxygen and four with controlled dissolved oxygen. Statistical differences between treatments were analyzed using *t*-test.

Treatment	Replicate Number	Batch Duration (Days)	Final Packed Cell Volume (%)	Specific Growth Rate (d^−1^)
Uncontrolled Dissolved Oxygen	1	19	85	0.089
	2	21	79	0.076
	3	19	70	0.078
	4	19	80	0.085
Controlled Dissolved Oxygen	1	9	70	0.164
	2	12	75	0.129
	3	10	90	0.173
	4	14	75	0.110
Statistical differences between treatments		*p* = 0.002	*p* = 0.85	*p* = 0.02

**Table 2 plants-14-02879-t002:** Measurements of volumetric mass transfer coefficient (*k*_L_*a*) in F0 and F2 under different process conditions. Results of three replicates using F0 (laboratory scale) with agitation speed of 80 rpm with 0.4 slpm aeration rate and 160 rpm with 1.2 slpm aeration rate, and F2 (pilot scale) of 80 rpm with 4 slpm aeration rate and 200 rpm with 8 slpm aeration rate. Temperature was constant at 25 °C. Pressure was constant at 0.05 barg.

Equipment	Working Volume (L)	Agitator Speed (rpm)	Aeration Rate (slpm)	Volume of CuSO_4_ 320 mg/L Solution (mL)	Total Mass Na_2_SO_3_ (g)	Net Mass Na_2_SO_3_ (g)	Test Duration (min)	Calculated Oxygen Transfer Rate (mmol/L-h)	Calculated Volumetric Mass Transfer Coefficient, *k*_L_*a* T (h^−1^)
F0	2	80	0.4	4	10	9.87	854.8	1.4	5.6
F0	2	80	0.4	4	10	9.87	729.7	1.6	6.6
F0	2	80	0.4	4	10	9.87	778.4	1.5	6.2
F0	2	160	1.2	4	10	9.87	238.5	4.9	20.2
F0	2	160	1.2	4	10	9.87	243.0	4.8	19.8
F0	2	160	1.2	4	10	9.87	239.3	4.9	20.1
F2	7	80	4	14	35	34.55	346.4	3.4	13.9
F2	7	80	4	14	35	34.55	403.2	2.9	11.9
F2	7	80	4	14	35	34.55	333.8	3.5	14.4
F2	7	200	8	14	35	34.55	115.0	10.2	41.9
F2	7	200	8	14	35	34.55	118.0	10.0	40.8
F2	7	200	8	14	35	34.55	114.3	10.3	42.1

**Table 3 plants-14-02879-t003:** Summary of critical parameters used for scale-up from laboratory-scale bioreactor Bionet F0-2CC to pilot-scale bioreactor Bionet F2-15MB for plant cell suspension cultivation.

Parameter	F0 (No Cascade)	F0 (Cascade Dissolved Oxygen Control)	F2 (Pilot Scale, Cascade Dissolved Oxygen Control)	Control
Working volume (L)	2.0	2.0	7.0	Maintained geometric similarity (Height/Diameter ratio)
Impeller diameter (mm)	54	54	70.5	Proportional
Agitation range (rpm)	Constant, 80	80–160	80–200	Adjusted via cascade control
Tip speed (m/s)	0.23	0.23–0.45	0.3–0.82	Adjusted to control *k*_L_*a*
Aeration rate (vvm)	0.2	0.2–0.6	0.6–1.1	Adjusted to control *k*_L_*a*
Dissolved oxygen setpoint (%)	None (natural decline)	30	30	Constant (cascade-based)
Average volumetric oxygen transfer coefficient, *k*_L_*a* (h^−1^)	~6.1	13.4–20.0	13.4–41.6	Targeted and matched
Reynolds number (*N*_RE_)	~373	580–910	580–910	Maintained hydrodynamic regime
Culture duration (days)	21	10	10	Significantly reduced
Final Packed Cell Volume (%)	80–90	~90	~80	Equivalent

## Data Availability

The original contributions presented in this study are included in the article. Further inquiries can be directed to the corresponding author.

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
