# Peer review of "Cascade Oxygen Control Enhances Growth of Nicotiana benthamiana Cell Cultures in Stirred-Tank Bioreactors"

_plants, 2025, doi:10.3390/plants14182879_

Round 1

Reviewer 1 Report

Comments and Suggestions for Authors

The manuscript (MS) entitled “Oxygen Transfer-Driven Optimization and Scale-Up of Nicotiana benthamiana Suspended Cell Cultures in Stirred-Tank Bio” is important for this species; however, the MS has serious omissions. It lacks of scientific novelty and is difficult to understand, I suggest providing the most interesting results. In general, a basic contribution for plant biotechnology. The manuscript requires major revisions:

-Keywords must be different from the title.

- The abstract must have an objective

- Lines 86 to 94 do not correspond to an introduction, it is required to include an objective.

- Results should not include methodology, for example, lines 97 to 102 are methodology. Only include outstanding results.

-Tables and figures should be more descriptive in their initials and treatments. These

should be self-explanatory.

- In figure 1 and 2, what do the two bottles mean?

- I didn't find any graph or table about the growth rate.

- Measurement parameters are required: growth curves, fresh weight, fresh weight, etc. Biochemical variables in response to treatments could be included.

- The discussion is very poor and should be improved (the authors should interpret, infer and analyze the results.  It is recommended to update references (2024-2025).

-Discussion part. It is not described what the authors consider to be the novelty of the presented work and its practical significance.

-What progress against the most recent state-of-the-art similar studies was made in this study? I strongly suggest highlighting these aspects in the introduction, discussion and conclusions.

For these reasons, I recommend that the manuscript cannot be accepted for publication in its current version.

Author Response

Referee 1

Thank you very much for your suggestions.

We are addressing your suggestions in the manuscript in red together with those from the other referees. We also answer your inquiries point-by-point below.

Comments and Suggestions for Authors

The manuscript (MS) entitled “Oxygen Transfer-Driven Optimization and Scale-Up of Nicotiana benthamiana Suspended Cell Cultures in Stirred-Tank Bio” is important for this species; however, the MS has serious omissions. It lacks of scientific novelty and is difficult to understand, I suggest providing the most interesting results. In general, a basic contribution for plant biotechnology. The manuscript requires major revisions:

>Thank you very much for your observation. We disagree with your view of lack of novelty because contrary to the extensive literature in bacteria, yeast and animal cells, very little has been published on the procedures for up-scaling plant cells in bioreactors. Furthermore, to the best of our knowledge, this is the first work where a cascade comprising both agitation and aeration has been described in detail. In our view, many aspects that may be considered common knowledge in fermentation have not been explicitly tested using plant cells. Nevertheless, as the frame of our contribution was somehow diffuse, we have increased the introduction and discussion including aspects from other referees. We would like to stress that obtaining a 2.4-fold increase in growth rate is not a trivial result.

-Keywords must be different from the title.

> The keywords have been changed according to your request.

- The abstract must have an objective.

>We have changed the abstract as requested.

- Lines 86 to 94 do not correspond to an introduction, it is required to include an objective.

>We have changed the paragraph accordingly.

- Results should not include methodology, for example, lines 97 to 102 are methodology. Only include outstanding results.

>We eliminated those sentences from Results as requested.

-Tables and figures should be more descriptive in their initials and treatments. These

should be self-explanatory.

>We have modified the figure legends as suggested.

- In figure 1 and 2, what do the two bottles mean?

>Thank you for your question. In both figures, the two bottles represent the PCV of the total culture harvested. We have written it in the figure legends.

- I didn't find any graph or table about the growth rate.

>Thank you very much for your observation. We did not take samples to measure growth rate as this modifies the actual working volume of the culture as this influences the geometry of the culture in an indirect way. Indeed, sampling 50 mL to measure PCV or Dry Cell Weight would result in a continuous reduction of the height of the culture compared to its width. This would be more critical in the bench-top bioreactor. Nevertheless, we have added a table with the growth rate calculated with the PCV as described in Materials and Methods.

- Measurement parameters are required: growth curves, fresh weight, fresh weight, etc. Biochemical variables in response to treatments could be included.

>Thank you for your comment. We have already answered this question above.

- The discussion is very poor and should be improved (the authors should interpret, infer and analyze the results.  It is recommended to update references (2024-2025).

> We have changed and rearranged the discussion. We added new references and subheadings. Please notice that relevant publications related to our work are not abundant so some of the new references are not from 2024-2025. Thank you very much.

-Discussion part. It is not described what the authors consider to be the novelty of the presented work and its practical significance.

> We have added new information in the discussion and added conclusions to improve this point. Thank you.

-What progress against the most recent state-of-the-art similar studies was made in this study? I strongly suggest highlighting these aspects in the introduction, discussion and conclusions.

>As previously, we have addressed these critics.

Reviewer 2 Report

Comments and Suggestions for Authors

The manuscript addresses an important topic with industrial relevance; however, it requires major revisions before it can be considered for publication. To ensure scientific rigor and clarity, the authors should provide additional methodological details, improve the presentation of the data with statistical analysis, strengthen the discussion with appropriate comparisons to the literature, and correct several language and formatting issues.

Detailed comments and suggestions are provided in the file below.

Author Response

Referee 2

The manuscript, entitled "Oxygen Transfer-Driven Optimization and Scale-Up of Nicotiana benthamiana Suspended Cell Cultures in Stirred-Tank Bioreactors", addresses a relevant topic in the field of plant bioprocess engineering. The manuscript proposes a dissolved oxygen (DO2) cascade control strategy, combining agitation and aeration. The authors report that maintaining DO₂ at 30% reduced culture time from 21 to 10 days and increased the specific growth rate more than twofold. Furthermore, the team asserts that they have successfully scaled up the process from a 2 L bench-top to a 7 L pilot bioreactor, while maintaining geometric similarity and key engineering parameters such as kLa and Reynolds number. While these findings have the potential to contribute to the field of process intensification, the study is not without its notable weaknesses. The methodological details are either missing or insufficiently described, the data presentation lacks statistical rigor, and some conclusions appear speculative or insufficiently supported by comparative literature. It is evident that, despite the topic's inherent interest and relevance within industrial contexts, substantial revisions are imperative to enhance the manuscript's strength and credibility. Detailed comments and suggestions are provided below.

Major Comments

  1. Provide more comprehensive information on the experimental design. Specify the number of replicates for all bioreactor runs and explain how variability was assessed.

>Thank you very much for the comment. Indeed, we have performed eight batches in the bench-top bioreactor, with and without cascade control. However, we have performed a single large-scale batch due to the difficulty to run large-scale experiments and the availability of the equipment. We have added a table with the corresponding data for the replicates.

  1. Please also include details of the statistical analyses performed (e.g., standard deviations, error bars, and significance tests). Currently, data are presented without variability estimates.

>We have added the statistical analysis of eight batches to address this comment. Thank you very much.

  1. In the case of bioreactor design, please describe the impeller design, including the number and angle of blades, and confirm whether the same type was used in both bioreactors.

>We have added the information related to the number of blades and their corresponding angle, which is the same for both bioreactors. Thank you.

  1. Clarify the gas composition used for aeration (air versus enriched oxygen) and whether antifoam agents were applied.

> Thank you very much. We used air. To clarify this point, we mentioned it in the manuscript (see line 372-373). We did not use antifoam agents in these trials; it was not necessary.

  1. Please, improve the readability of Tables 1 and 2 by adding clear column headings with units and including measures of variability. Figures referenced in the text (e.g., DO₂, pH, and PCV profiles) also require clear legends and units. Ensure that colors and labels are visible and consistent.

> We have changed the headings, legends and units of the figures and legends to improve the readability of them. Thank you so much for your comment.

  1. Some claims in the discussion section are speculative. For instance, the statement that the process can be scaled up to 100–500 litres lacks support from experiments or literature. This should be toned down or supported with appropriate citations.

> Thank you for the comment. We have included new literature supporting the statement, but we have also toned it down.

  1. Please also discuss the potential limitations of your study, such as its applicability to different plant cell lines and its effect on product yield rather than just biomass.

>We have remarked the limitations of our study in the discussion and conclusions. Thank you very much.

  1. Compare your kLa and µ values explicitly with those reported in recent literature for N. benthamiana and other plant species, in order to strengthen your findings.

> Thank you very much for your comment, that was already mentioned in the document lines 267-277 for µ and 288-299 for kLa.

  1. Update the introduction with more recent references (2023–2025) that are relevant to DO₂ control and the scaling up of plant cell culture in various kinds of bioreactors (for example, based on this literature: https://doi.org/10.3389/fbioe.2024.1461253, https://doi.org/10.1038/s41598-025-92385-y, https://doi.org/10.1016/j.cej.2024.155966, https://doi.org/10.3390/bioengineering12040332, https://doi.org/10.1111/pbi.70153, https://doi.org/10.1016/j.tibtech.2023.09.003).

> Thank you very much for the examples. We have added the relevant papers.

  1. Strengthen the claim that 'there are no reports describing cascade control with both aeration and agitation' by conducting a thorough literature review.

>Indeed, experiments have evaluated either increased aeration or increased agitation but not both in a coordinated fashion. We have added the corresponding publications in the introduction and discussion. Thank you.

  1. Consider shortening or refining the title to focus on the main contribution (DO₂ cascade control and scale-up).

>Thank you so much. We have changed the title to: Cascade Oxygen Control Enhances Growth of Nicotiana benthamiana Cell Cultures in Stirred-Tank Bioreactors.

Minor Comments

  1. In the abstract, include specific quantitative results (e.g., exact µ values and kLa ranges) to highlight the key findings more effectively.

> We have included the specific quantitative results in the abstract following your suggestion. Thank you.

  1. Clarify whether pH was controlled or only monitored and discuss the potential effect of pH changes on growth.

> We clarified that pH was monitored but was not controlled in the manuscript. Thank you.

  1. Use consistent notation for µ (specific growth rate) and units (always use “d⁻¹” rather than mixing with “d-1”, for example).

> We checked for the notation for µ and d-1. Thank you very much.

  1. Revise the manuscript for grammar, spelling, and style consistency. Examples include “Emrbyonic” (should be “Embryonic”), “his scale” (should be “this scale”), and inconsistent unit formatting (e.g., “0.2vvm” vs. “0.2 vvm”).

> We have revised the manuscript for these concerns. Thank you very much for your comment.

The manuscript addresses an important topic with industrial relevance; however, it requires major revisions before it can be considered for publication.

To ensure scientific rigor and clarity, the authors should provide additional methodological details, improve the presentation of the data with statistical analysis, strengthen the discussion with appropriate comparisons to the literature, and correct several language and formatting issues

>We have included additional statistics and increased discussion and literature comparisons. We have professional English editors thanks

Reviewer 3 Report

Comments and Suggestions for Authors
  1. In academic writing, dissolved oxygen is generally abbreviated as “DO”; “DO2” is prone to ambiguity.
  2. Lines 66–68: What do the following parameters represent—volumetric oxygen transfer coefficient, power input per unit volume, Reynolds number, aeration rate, impeller tip speed, and mixing time? Is a larger value better or a smaller one? Non-specialist readers may not understand this.
  3. Lines 75–78: Please add examples of different bioreactors used for plant cell culture to make the argument more convincing.
  4. Lines 86–94: In the final paragraph of the introduction, I would prefer to see what method this study used to address N. benthamianacell production, rather than presenting results.
  5. Line 99: There is no space between numbers and units; please search and correct throughout the manuscript. In “°C,” why is there a horizontal line beneath the degree symbol?
  6. Section 2.1: The subheading is not appropriate. This section monitors changes in various indicators during bioreactor culture with and without cascade control. In academic papers, subheadings generally should not state conclusions directly; please revise. The subheadings of Sections 2.2 and 2.3 are also inappropriate; please revise.
  7. Table 1: “OTR” does not appear earlier in the text; please indicate what it is an abbreviation for.
  8. Table 2: The last column header should not be “Scale-Up / Control Criterion.” “Scale-Up / Control Criterion” should be grouped with “Parameter,” while the last column describes how each parameter is controlled or its characteristics. Consider renaming the last column to “Control” or “Characteristics.”
  9. Figure 3A: Why does agitation drop at 215–225 hours and then recover? Please explain this in the text.
  10. The discussion contains few comparative examples; consider dividing it into several subheadings.
  11. Line 261: Fix the subscripts for monopotassium phosphate.
  12. Methods: There is no statistical analysis—how were the figures produced?
  13. Please add a conclusion section that summarizes the manuscript.
  14. References: Punctuation is incorrect; some entries lack page numbers. If journal names are abbreviated, abbreviate all of them consistently.

Author Response

  1. In academic writing, dissolved oxygen is generally abbreviated as “DO”; “DO2” is prone to ambiguity.
  • Thank you for your suggestion, we have changed the abbreviation.
  1. Lines 66–68: What do the following parameters represent—volumetric oxygen transfer coefficient, power input per unit volume, Reynolds number, aeration rate, impeller tip speed, and mixing time? Is a larger value better or a smaller one? Non-specialist readers may not understand this.
  • Thank you for your comment. We added explanations for each of the parameters.
  1. Lines 75–78: Please add examples of different bioreactors used for plant cell culture to make the argument more convincing.
  • Thank you for your comment. We have expanded part including examples on rocking bioreactors, using some of the suggested references.
  1. Lines 86–94: In the final paragraph of the introduction, I would prefer to see what method this study used to address N. benthamianacell production, rather than presenting results.
  • Thank you for the suggestion. We have changed the last paragraph.
  1. Line 99: There is no space between numbers and units; please search and correct throughout the manuscript. In “°C,” why is there a horizontal line beneath the degree symbol?
  • Thank you for your comment. We have added space between numbers and units, and we have also changed the ºC to not have that horizontal line.
  1. Section 2.1: The subheading is not appropriate. This section monitors changes in various indicators during bioreactor culture with and without cascade control. In academic papers, subheadings generally should not state conclusions directly; please revise. The subheadings of Sections 2.2 and 2.3 are also inappropriate; please revise.
  • Thank you for the suggestion. We have changed the subheadings.
  1. Table 1: “OTR” does not appear earlier in the text; please indicate what it is an abbreviation for.
  • Thank you for the comment. We have put directly the definition instead of the abbreviation.
  1. Table 2: The last column header should not be “Scale-Up / Control Criterion.” “Scale-Up / Control Criterion” should be grouped with “Parameter,” while the last column describes how each parameter is controlled or its characteristics. Consider renaming the last column to “Control” or “Characteristics.”
  • Thank you for your suggestion. We have renamed the column as “Control”.
  1. Figure 3A: Why does agitation drop at 215–225 hours and then recover? Please explain this in the text.
  • Thank you for your question. We have added the explanation related to increased aeration and decreased agitation.
  1. The discussion contains few comparative examples; consider dividing it into several subheadings.
  • We have divided the discussion as suggested and added comparative examples. Thank you.
  1. Line 261: Fix the subscripts for monopotassium phosphate.
  • Thank you for your suggestion. We have fixed it.
  1. Methods: There is no statistical analysis—how were the figures produced?
  • Thank you for the comment. The figures are produced with the csv data obtained from bioreactor software and using ggplot2. We have added a small part in material and methods.
  1. Please add a conclusion section that summarizes the manuscript.
  • We have added a conclusions section. Thank you so much.
  1. References: Punctuation is incorrect; some entries lack page numbers. If journal names are abbreviated, abbreviate all of them consistently.

We have checked the references and changed the ones with errors in them. Thank you for your comment

Round 2

Reviewer 1 Report

Comments and Suggestions for Authors

The manuscript (MS) entitled: “Cascade Oxygen Control Enhances Growth of Nicotiana benthamiana Cell Cultures in Stirred-Tank Bioreactors, has important changes; however, the following points were not addressed:

-Tables and figures should be more descriptive in their initials and treatments. These should be self-explanatory.  Abbreviations must be described in all tables and figures, example in Table 1: PCV, F0-2 CC, DO, μ (d-1).

Author Response

Referee 1-round-2

The manuscript (MS) entitled: “Cascade Oxygen Control Enhances Growth of Nicotiana benthamiana Cell Cultures in Stirred-Tank Bioreactors, has important changes; however, the following points were not addressed:

-Tables and figures should be more descriptive in their initials and treatments. These should be self-explanatory.  Abbreviations must be described in all tables and figures, example in Table 1: PCV, F0-2 CC, DO, μ (d-1).

  • Thank you for your comment. We have added the abbreviations in all tables and figures.

Reviewer 2 Report

Comments and Suggestions for Authors

The authors have addressed all major and minor comments from my previous review. The revised manuscript now includes improved methodological detail, statistical analysis, updated references, and a more balanced discussion. I recommend this manuscript for publication in Plants.

Author Response

Ref2-v2

Thanks for recommendation for publication

Reviewer 3 Report

Comments and Suggestions for Authors

This study employed tobacco suspension cells as the research model and cultivated them in bioreactors using a Nicotiana benthamiana dissolved oxygen control strategy that combined stirring and aeration. Compared with non-controlled conditions, the specific growth rate increased by more than twofold, the cell volume fraction rose from approximately 16% to about 80%, and the time required to reach high biomass was shortened from 21 days to 10 days. The process was successfully scaled up from a 2 L benchtop bioreactor to a 7 L stainless steel bioreactor, and reproducibility was achieved based on geometric similarity and mass transfer–flow principles. These results demonstrate that conventional equipment can be effectively applied to markedly enhance the efficiency of plant cell cultures.This manuscript has both strengths and limitations.

Weaknesses:

The manuscript lacks sufficient detail in certain aspects. For example, it frequently refers to “inoculated at 20%,” “packed cell volume adjusted to 16%,” or “PCV of 16%,” but does not specify how PCV was measured. The sterilization method is not described, nor is it explained how probes and tubing were adequately sterilized. In addition, the number of replicates for each experimental condition is not indicated. These details are recommended for inclusion.

Author Response

The manuscript lacks sufficient detail in certain aspects. For example, it frequently refers to “inoculated at 20%,” “packed cell volume adjusted to 16%,” or “PCV of 16%,” but does not specify how PCV was measured.

  • Thanks for the comment. The reason is the following: we described with detailed on lines 333, 339, 349 and 354 that we inoculate at 20% v/v ratio. When you calculate the final PCV, if the inoculum had a PCV of 80%, you end up with 16% PCV, as initial biomass. PCV measurement is described in lines 399 and 400.

The sterilization method is not described, nor is it explained how probes and tubing were adequately sterilized.

  • Thank you for your comment. We have added more detailed information about the sterilization processes.

In addition, the number of replicates for each experimental condition is not indicated. These details are recommended for inclusion.

  • In the prior revision we included the number of replicates in lines 113-114. Thank you very much.
